**Data Availability Statement:** All relevant data are within the paper and its Supporting Information files.

# An updated systematic review and meta-analysis on efficacy of Sofosbuvir in treating hepatitis C-infected patients with advanced chronic kidney disease

Sara Majd Jabbari[1], Khadije Maajani[2], Shahin Merat[3], Hossein Poustchi🅾[3], Sadaf G. Sepanlou🅾[1]*

**1** Digestive Diseases Research Center, Digestive Diseases Research Institute, Tehran University of Medical Sciences, Tehran, Iran, **2** Department of Epidemiology and Biostatistics, School of Public Health, Tehran University of Medical Sciences, Tehran, Iran, **3** Liver and Pancreatobiliary Diseases Research Center, Digestive Diseases Research Institute, Tehran University of Medical Sciences, Tehran, Iran

* sepanlou@yahoo.com, sgsepanlou@tums.ac.ir

## Abstract

Sofosbuvir seems to be a revolutionary treatment for Hepatitis C-infected patients with advanced chronic kidney disease (CKD) but existing evidence is not quite adequate. The aim of this study was to evaluate the efficacy and safety of Sofosbuvir-based therapy without Ribavirin for all hepatitis C virus genotypes among patients with advanced CKD. We conducted an updated systematic literature search from the beginning of 2013 up to June 2020. Sustained virologic response (SVR) rate at 12 and/or 24 weeks after the end of treatment, and adverse events in HCV-infected patients with advanced CKD were pooled using random effects models. We included 27 published articles in our meta-analyses, totaling 1,464 HCV-infected patients with advanced CKD. We found a substantial heterogeneity based on the $I^2$ index (P = 0.00, $I^2$ = 56.1%). The pooled SVR rates at 12 and 24 weeks after the end of Sofosbuvir-based treatment were 97% (95% Confidence Interval: 95–99) and 95% (89–99) respectively. The pooled SVR12 rates were 98% (96–100) and 94% (90–97) in patients under 60 and over 60 years old respectively. The pooled incidence of severe adverse events was 0.11 (0.04–0.19). The pooled SVR12 rate after completion of the half dose regimen was as high as the full dose treatment but it was associated with less adverse events (0.06 versus 0.14). The pooled SVR12 rate was 98% (91–100) in cirrhotic patients and 100% (98–100) in non-cirrhotic patients. The endorsement of Sofosbuvir-based regimen can improve the treatment of hepatitis C virus infection in patients with advanced CKD.

## Introduction

Chronic kidney disease (CKD) is described as the gradual loss of kidney function over time. CKD progresses through five stages [1]. Previous reports reveal that chronic hepatitis C virus (HCV) infection is the most common chronic liver disease in patients with advanced CKD [2].

**Funding:** The authors received no specific funding for this work.

**Competing interests:** The authors have declared that no competing interests exist.

The prevalence varies worldwide, with a higher proportion of infected patients in developing countries than in developed nations [3, 4]. In Western countries, it has been estimated that almost 6% of patients with advanced CKD, who are on conservative therapy, are infected with HCV [5]. Approximately 4% to 70% of patients on hemodialysis (HD), and between 11% and 49% of kidney transplant (KTx) recipients are infected with HCV [6]. HCV infection is detected as a frequent cause of morbidity, mortality, and graft failure among patients on HD and kidney transplant (KTx) recipients [7–10].

In earlier studies, PEGylated interferon (Peg-IFN) monotherapy showed higher SVR rates (approximately 40%) compared to conventional interferon therapy (approximately 30%) in HCV-infected CKD patients [11]. The addition of ribavirin (RBV) did not result in higher SVR rates compared to regimens restricted to peg-IFN. Antiviral therapy with peg-IFN with or without RBV was not quite effective and was associated with several side effects, which raised concerns about its safety [12]. Interferon and RBV are associated with significant toxicity including anemia, depression, anxiety, and other psychiatric side effects [13–15]. As a result, the main limitation of interferon-based regimens is their high withdrawal rates due to side effects (26.9%) [12].

Sofosbuvir is classified as a superior direct-acting nucleotide polymerase inhibitor that inhibits HCV replication through inhibition of NS5B RNA polymerase, which regulates the chronic HCV infection. This drug is metabolized in vivo into its active intracellular metabolite GS-461203 and the pharmacologically inactive form GS-331007. This active triphosphate of Sofosbuvir, GS-461203, targets the vastly conserved active site of the HCV-specific NS5B polymerase and acts as a nonobligatory chain terminator, an effect that is independent of the viral genotype [16]. The use of Sofosbuvir (400 mg/day) leads to drastically increased level of inactive metabolite GS331007 in patients with e-GFR $< 30$ mL/min/1.73 $m^2$, which raises concerns about the accumulation of GS331007 in patients with advanced CKD, possibly leading to toxicity as GS331007 is mostly eliminated by kidneys [17, 18]. The effective and safe dose of Sofosbuvir in advanced CKD patients is thus not established. Still, existing evidence on usage of Sofosbuvir in patients with e-GFR$< 30$ ml/min/1.73 $m^2$ is accumulating [19].

In a recent systematic review, the safety and efficacy of Sofosbuvir was explored among patients on HD [20] and results are quite similar to the results of the current study. However, there is a very important difference between the two studies. In our study, we included patients with eGFR$<30$ ml/min/1.73 $m^2$ and our search was not confined to patients on HD. This difference is quite important as the risk of toxic metabolite accumulation in patients with advanced CKD and not necessarily on HD may be even higher than patients on HD. We also excluded studies on acute HCV and studies on patients with HBV and HIV co-infection and we conducted subgroup and sensitivity analyses. Thus, in this updated systematic review and meta-analysis we used the currently available evidence to evaluate the antiviral efficacy of SOF-based therapy in combination with other direct acting antivirals (DAA) in advanced CKD patients infected chronically with HCV.

## Materials and methods

We performed this systematic review to identify studies on Sofosbuvir-based antiviral therapy among HCV-infected patients with advanced CKD. This study was conducted in accordance with preferred reporting items for systematic reviews and meta-analyses (PRISMA) guidelines [21].

### Search strategy

For systematic literature search, we used several international databases including PubMed/Medline, Web of sciences, Scopus and CENTERAL on the Cochrane library from January 1st,

2013 (the year Sofosbuvir was approved for medical use) up to June 26[th], 2020. The search strategy was based on a combination of key words from medical subject headings (Mesh) and free texts including "(dialysis OR hemodialysis OR end stage renal disease OR chronic kidney failure OR chronic kidney disease OR severe renal impairment) AND (Sofosbuvir OR N5B polymerase OR direct-acting antiviral). The details of our search strategy are presented in the S1 Appendix in S1 File. Finally, to avoid missing any relevant evidence, we reviewed the reference list of all included studies.

## Study selection

After literature search by two independent reviewers (Kh M and S M), results of the initial search were incorporated into the reference manager software, duplicates were removed, and the two reviewers screened the studies by title, abstract, and full text independently and included or excluded them based on defined eligibility criteria.

## Eligibility criteria

All studies would be eligible if they were published in full English and reported the efficacy and safety profiles of Sofosbuvir-based regimen for treatment-naive chronic HCV-infected patients (aged $\geq$ 18 years) with advanced CKD, defined as e-GFR $\leq$ 30 ml/min per 1.73 $m^2$ or being on dialysis. Studies on all HCV genotypes were included. Outcomes of interest were SVR rates at 12 or 24 weeks after the end of treatment. Studies were excluded if there was no sufficient data to calculate SVR12 or SVR24 rate and if HCV-infected patients were co-infected with other viral infections such as hepatitis B and HIV/AIDS.

## Quality assessment

To assess the quality of included studies we used Newcastle—Ottawa quality assessment scale for cohort studies, which comprised three sections as follow: 1. Selection, 2. Comparability, and 3. Outcome [22]. Therefore, studies with cumulative score equal to 7 or more were classified as high quality studies (n = 6) and studies with cumulative scores between 4 and 6 were defined as fair quality studies (n = 21). Two reviewers (Kh M and S M) independently assessed the quality of each study. In order to explore the agreement between the two reviewers, we used Kappa statistics (Weighted kappa = 87%). Disagreements between the reviewers were resolved through discussions, or the final decision was made based on the opinion of the third reviewer (SM).

## Data extraction

To extract the relevant data, two reviewers (Kh M and S M) independently used a pre-specified data extraction sheet in Microsoft Excel, which consisted of the following variables: name of the first author, year of publication, time of the study, country, study design, number of patients, mean age, sex distribution, baseline HCV RNA level, cirrhosis diagnosis, treatment strategy, doses of drugs and duration of treatment, the rate of SVR12 and/or SVR24, side effects, and mortality rate.

## Statistical analysis

To investigate the between study heterogeneity we used $I^2$ index and chi-squared test at the 5% significance level (p <0.05). A random effects model was used to pool the SVR12 rate. We used metaprop command to estimate the exact binomial and score-test-based confidence intervals for proportions near boundaries (i.e., near 100% or zero) [23]. We also conducted

subgroup analysis and meta-regression by treatment strategy, dose of Sofosbuvir, diagnosis of cirrhosis, country of origin, and age. In this study, publication bias was not assessed, as the SVR rate is a proportion, is always a positive number, and the asymmetry in the funnel plot is not due to publication bias. This meta-analysis was performed in Stata 11 (StataCorp, College Station, TX, USA).

## Results

### Study characteristics

Our final systematic search yielded 27 relevant articles based on the eligibility criteria and we included these studies in our systematic review and meta-analyses (Fig 1). These studies were conducted in 10 countries, comprising 6 studies in the USA, 10 in India, 2 in France, 2 in Pakistan and single studies in Austria, Czech Republic, Iran, Korea, Egypt, and Brazil (Table 1). Detailed study characteristics are demonstrated in S1 Table in S1 File. The included studies contained extractable data on 1464 patients, among whom 809 were males and 655 were females. The mean age of all enrolled patients was 48.6±11 year. We included all articles that used Sofosbuvir-based therapies, and the enrollment period of these patients ranged from 2013 to 2020. All patients in 22 studies received full dose Sofosbuvir (400 mg) daily, while patients in 5 studies received half dose (200mg) Sofosbuvir daily. In all of the studies, Sofosbuvir was used in combination with other direct-acting antivirals (DAAs). Among all patients enrolled in the included studies, only 274 patients were cirrhotic. Characteristics of included studies are demonstrated in Table 1.

### Heterogeneity

The between study heterogeneity based on $I^2$ index (P = 0.00, $I^2$ = 56.1%) was substantial.

### Pooled SVR rate

Based on the random effects model, the pooled SVR12 and 24 rates were 97% (95% CI: 95–99) and 95% (89–99) respectively in our meta-analysis (Fig 2 and S1 Fig in S1 File).

### Subgroup analysis

According to the results of subgroup analysis in Table 2, the pooled SVR12 rates were 98% (96–100) and 94% (90–97) in patients under 60 and over 60 years old respectively (S2 Fig in S1 File). The pooled SVR12 rate was 98% (91–100) in patients with cirrhosis and 100% (98–100) in non-cirrhotic patients (Fig 3). In subgroup analysis based on Sofosbuvir dose, the pooled SVR12 rate was 97% (94–99) in studies using full dose regimen (400 mg), and 99% (91–100) in studies using half dose (200mg) regimen (Fig 4). Results sub-grouped by region of study are presented in S3 Fig in S1 File showing lower SVR12 rates in Europe [95% (89–100)]. Fig 5 demonstrates the pooled SVR12 rate sub-grouped based on being treated only with Sofobuvir [99% (98–100)] or its combination with RBV [99% (95–100)]. We additionally defined subgroups by treatment strategy such as the result of the pooled SVR12 rate in 19 studies in which Sofosbuvir was used in combination with Daclatasvir [97% (94–99)], Simeprevir [99% (94–100)], and Ledipasvir [100% (100–100)]. Detailed results are reported in Fig 6. Table 2 demonstrates the summary of all sub-group analyses.

### Safety

Severe adverse events (SAE) were reported in 6 studies (S4 Fig in S1 File). The pooled incidence of SAE was 0.11 (0.04–0.19), and the incidence of SAE in patients who received full dose SOF compared to half dose were 0.14 (0.04–0.28) vs. 0.06 (0.01–0.15). Mortality was reported in 11 studies, and none of them were due to treatment (S5 Fig in S1 File). Therefore, the

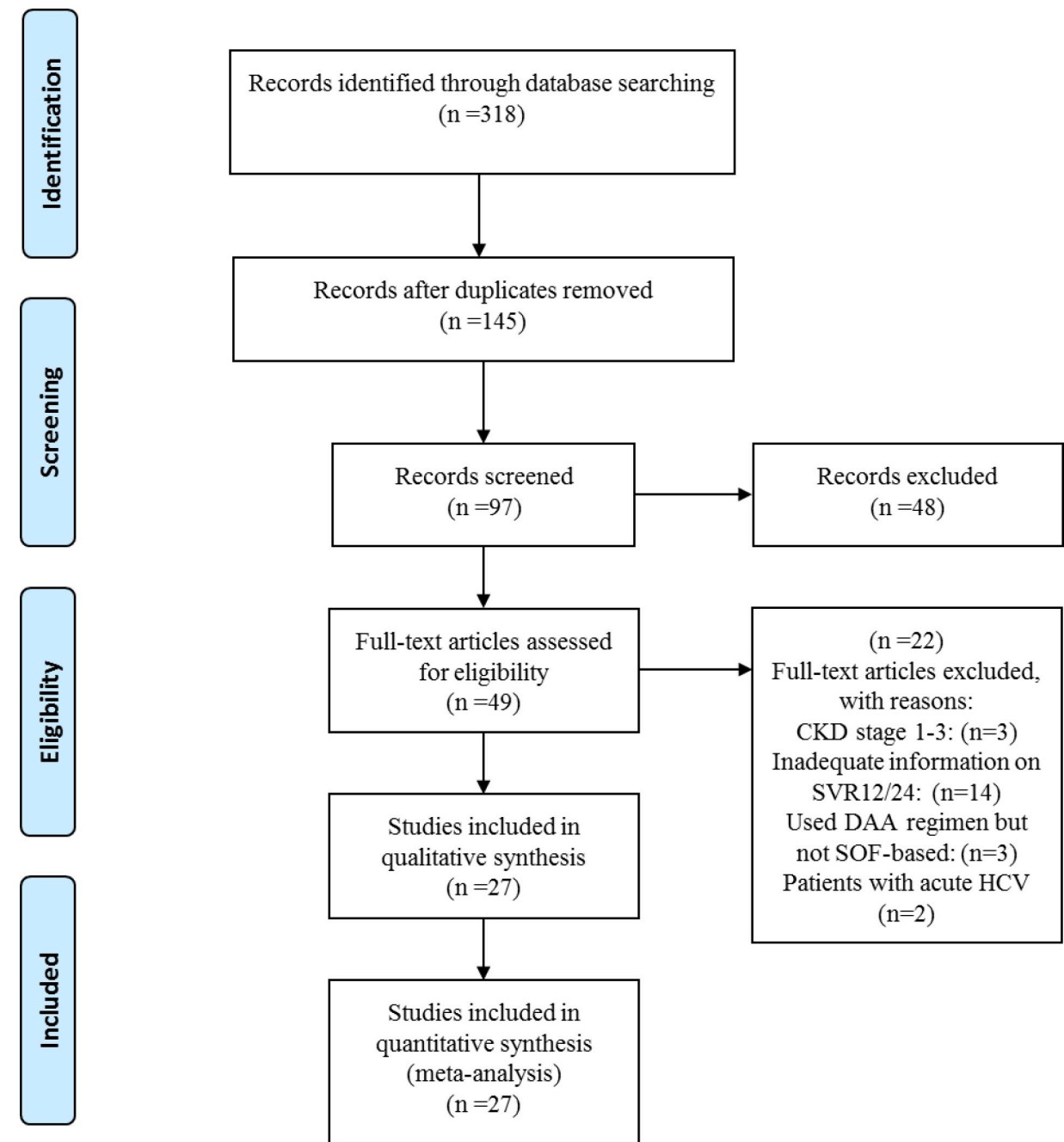

**Fig 1. PRISMA flowchart showing different phases of selecting relevant publications.**

estimated pooled mortality rate was 0.04 (0.01–0.09). In eight studies the discontinuation rate was reported and the pooled rate was 0.04 (0.00–0.11). (S6 Fig and S2 Table in S1 File)

### Meta-regression analysis

The results of meta-regression analysis are shown in Table 3. There was a significant association between age (P = 0.03, β = 13.4) and country (P = 0.02, β = -18.1) and the pooled SVR12

**Table 1. Study characteristics included in meta-analysis of SVR12 rate among HCV-infected patient with advanced chronic kidney disease.**

| First author/ year | Country | No. of patients | Recruitment period | Mean Age (SD) | Cirrhosis N (%) | No. of patient on dialysis/Total patient | SVR12 | SVR24 | NOS score |
|---|---|---|---|---|---|---|---|---|---|
| Taneja 2018 [24] | India | 65 | 2016 | 49 (13) | 21 (32.3%) | (54/65) | 100 | _ | 6 |
| Surendra 2018 [25] | India | 19 | 2016 | 44 | 0 | (19/19) | 100 | _ | 6 |
| Manoj 2018 [26] | India | 71 | 2015–2017 | 42 | 17 (23.9%) | (11/71) | 100 | 98.5 | 6 |
| Akhil 2018 [16] | India | 22 | 2015–2016 | 49.8 | 0 | (22/22) | 100 | _ | 6 |
| Sperl 2017 [27] | Czech Republic | 6 | 2015–2016 | 39 | 2 (33.3%) | (6/6) | 100 | _ | 5 |
| Dumortier 2017 [28] | France | 50 | 2013–2015 | 60.5 (7.5) | 0 | (35/50) | 86 | _ | 6 |
| Cox-North 2017 [29] | USA | 29 | 2014–2016 | _ | 13 (44.8%) | (0/29) | 97 | _ | 6 |
| Choudhary 2017 [30] | India | 16 | 2015–2016 | 45 (12) | 2 (12.5%) | (16/16) | 80 | _ | 6 |
| Aggarwal 2017 [31] | USA | 14 | _ | 61 (4.9) | 12 (85.7%) | (14/14) | 92.8 | _ | 6 |
| Agarwal 2017 [32] | India | 62 | 2015–2016 | 33.8 (10.2) | 3 (4.8%) | (62/62) | 95.2 | _ | 6 |
| Singh 2016 [33] | USA | 8 | 2014–2015 | 56.8 (20) | 3 (37%) | (8/8) | 100 | _ | 6 |
| Saxena 2016 [34] | USA | 18 | _ | ≥65 | - | - | 88 | _ | 8 |
| Nazario 2016 [35] | USA | 17 | _ | 57 | 8 (47%) | (17/17) | 100 | _ | 6 |
| Desnoyer 2016 [36] | France | 12 | 2014–2015 | 52 | 10 (83.4%) | (12/12) | 83.3 | 100 | 6 |
| Beinhardt 2016 [37] | Austria | 10 | _ | 50.6 (10.9) | 4 (40%) | (10/10) | 96 | 96 | 8 |
| Hundemer 2015 [38] | USA | 6 | 2014 | 60 (14) | 3 (50%) | (6/6) | 64 | _ | 6 |
| Goel 2018 [39] | India | 41 | 2015–2017 | 41 | 5 (12%) | (41/41) | 90.2 | _ | 6 |
| Gupta 2018 [40] | India | 7 | 2015–2016 | 48.8 ± 14.5 | 2 (28.6%) | (7/7) | 100 | _ | 6 |
| Mehta 2018 [41] | India | 38 | 2016 | 49.5 | _ | (26/26) | 100 / 92.3 | _ | 6 |
| Borgia 2019 [42] | Canada, the United Kingdom, Spain, Israel, New Zealand, and Australia | 59 | 2017–2018 | 60 | 17 (29%) | (59/59) | 95 | _ | 6 |
| Poustchi 2020 [43] | Iran | 103 | 2017–2018 | 50.3±13.5 | 39 (37.9%) | (75/103) | 100 | _ | 6 |
| Eletreby 2020 [44] | Egypt | 579 | 2014–2018 | 52 | 107 (11%) | 10/579 | 96.7 | _ | 8 |
| Debnath 2020 [45] | India | 18 | 2017–2018 | 39.4 ± 8.3 | 0 | 18/18 | 100 | _ | 6 |
| Michels 2020 [46] | Brazil | 34 | 2016–2017 | 60.7±10.4 | 0 | 34/34 | 99.3 | 97.1 | 8 |
| Cheema 2019 [47] | Pakistan | 18 | 2017–2018 | 47.2±14.1 | 4 (22.2%) | 18/18 | 83.3 | 83.3 | 8 |
| Mandhwani 2020 [48] | Pakistan | 133 | 2016–2018 | 31.9 ± 9.8 | 0 | 133/133 | 96.9 | _ | 8 |
| Seo 2020 [49] | Korea | 9 | 2017–2018 | 59.9 | 2 (22.2%) | 9/9 | 100 | _ | 6 |

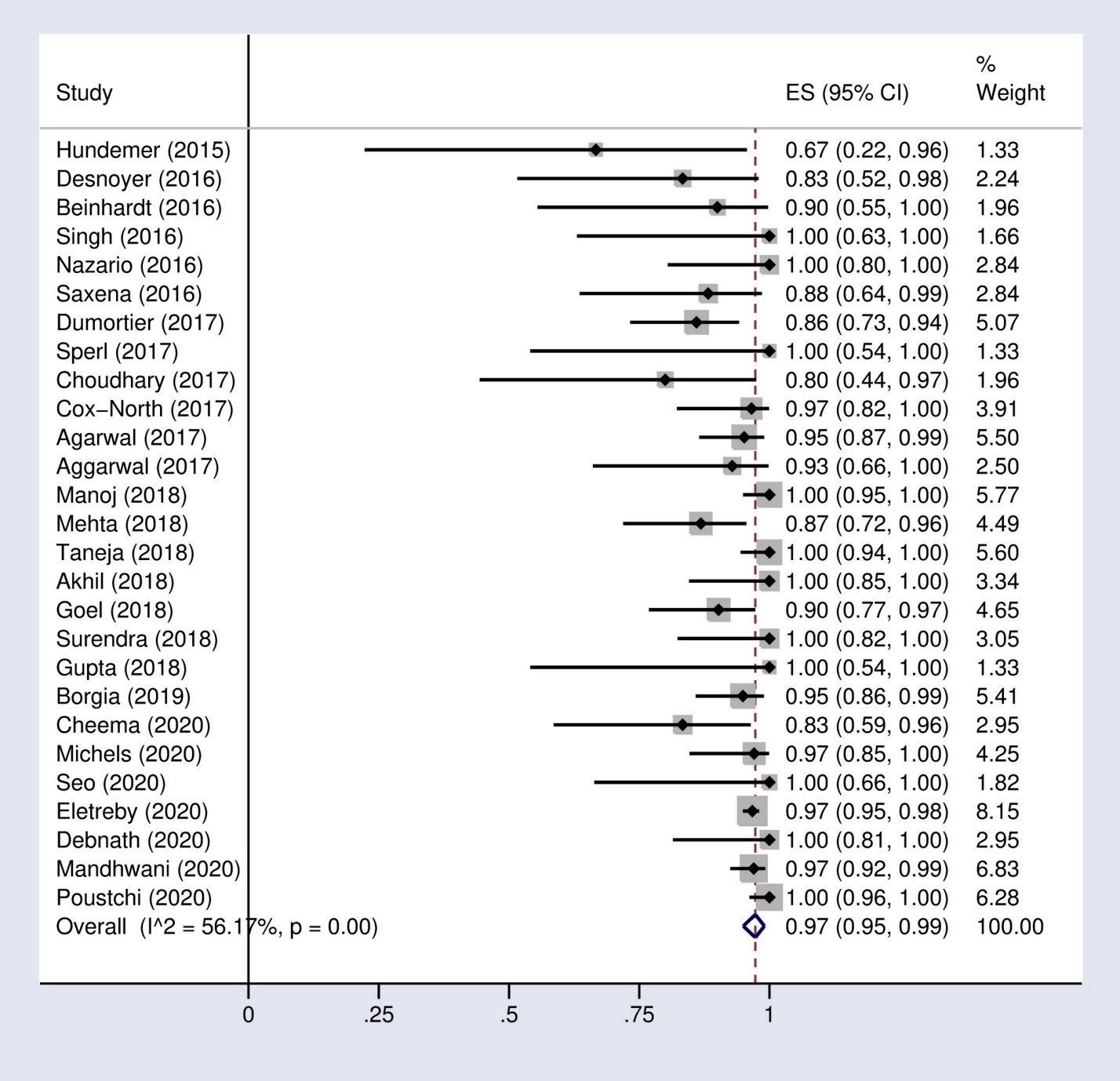

**Fig 2. Forest plot of the pooled SVR12 rate in HCV-infected patients with advanced chronic kidney disease.**

rate in the univariable model, which disappeared in the multivariable model. None of the variables of cirrhosis diagnosis, treatment strategy, and dose of treatment had significant association with the pooled SVR12 rate, neither in the univariable model nor in the multivariable meta-regression model.

**Table 2. Results of subgroup analysis for estimating the pooled SVR12 rate in HCV-infected patients with advanced CKD.**

|   | Title | Subgroup | Number of included studies | SVR12 | 95%CI | $I^2$ (P) |
|---|-------|----------|---------------------------|-------|-------|-----------|
| 1 | Treatment strategy | SOF+DCV | 19 | 97% | (94–99) | 53.6% (P = 0.00) |
|   |  | SOF+SMV | 9 | 99% | (94–100) | 0.00 (P = 0.83) |
|   |  | SOF+LDV | 6 | 100% | (100–100) | 0.00 (P = 0.99) |
|   |  | SOF+RBV | 8 | 97% | (83–100) | 57.6% (P = 0.02) |
|   |  | SOF+DCV +RBV | 3 | 97% | (95–99) | 0.00 (P = 0) |
|   |  | SOF+PEG+RBV | 3 | 96% | (79–100) | 0.00 (P = 0) |
| 2 | Type of combination | Only SOF-based | 27 | 99% | (98–100) | 29.5% (P = 0.06) |
|   |  | SOF-based + RBV | 11 | 99% | (95–100) | 51.8% (P = 0.01) |
| 3 | Dose of treatment | Full dose (400 mg) | 22 | 97% | (94–99) | 58.4% (P = 0.00) |
|   |  | Half dose (200 mg) | 5 | 99% | (91–100) | 53.7% (P = 0.07) |
| 3 | Diagnosis of cirrhosis | Cirrhotic | 12 | 98% | (91–100) | 54.7% (P = 0.01) |
|   |  | Non-Cirrhotic | 19 | 100% | (98–100) | 38.7% (P = 0.04) |
| 4 | Age group | <60 year | 19 | 98% | (96–100) | 60.1% (P = 0.00) |
|   |  | >60 year | 8 | 94% | (90–97) | 0.00 (P = 0.57) |
| 5 | Region | America | 7 | 96% | (91–100) | 16.1% (P = 0.31) |
|   |  | Asia | 15 | 98% | (94–100) | 68.7% (P = 0.00) |
|   |  | Europe | 4 | 95% | (89–100) | 0.00 (0.5) |

†SVR = Sustained viralogic response. RBV = Ribavirin. SOF = Sofosbuvir. DCV = Daclatasvir. LDV = Ledipasvir. SMV = Simeprevir.

## Sensitivity analysis

To assess the effect of all studies on the pooled SVR12 rate, we used sensitivity analyses. In each step, we excluded one of the total of 27 studies and calculated the pooled SVR12 rate across the remaining 26 studies. Thus, we performed 27 sensitivity analyses and we made 27 different estimates of SVR12 upon exclusion of each study in each step. The highest and lowest estimates after excluding the studies by Dumortier [28] and Poustchi [43] were 98.9% (90.8–100) and 95.3% (91.1–100) respectively. So the pooled SVR12 rate was stable across studies. Our results are actually robust and don't depend on the choice of included studies.

## Discussion

HCV-infected patients with advanced CKD form a susceptible population that can be treated with a combination of Sofosbuvir-based regimen and antiviral therapy. We included 27 relevant studies, comprising 1464 patients in our meta-analysis. The pooled SVR12 rate was 97%, which was an acceptable outcome.

The endorsement of Sofosbuvir-based regimen was a major advance in treatment of HCV-infected patients with stage 4–5 of CKD. Efficacy and safety profile concurs with high rates of SVR12-24 with few side effects.

In the systematic review conducted by Li M et al in 2019, the pooled SVR12/24 rate (97.1%) achieved by the SOF-based regimen was compatible with the results of our meta-analysis [50]. Also in the study conducted in 2016 by Li T et al [51], the reported SVR12 rate of DAAs-based regimens was 93.2%, which was lower compared to the result of our study as we focused only on Sofosbuvir-based regimen. In comparison with studies that included non-SOF-based therapies, our study revealed higher SVR12 rates and better tolerability following SOF-based therapy in HCV-infected patients with advanced CKD.

To assess the safety of SOF-based therapy, we computed the pooled discontinuation rate due to adverse events, which was just 4%. The low estimate reveals the satisfactory outcome of

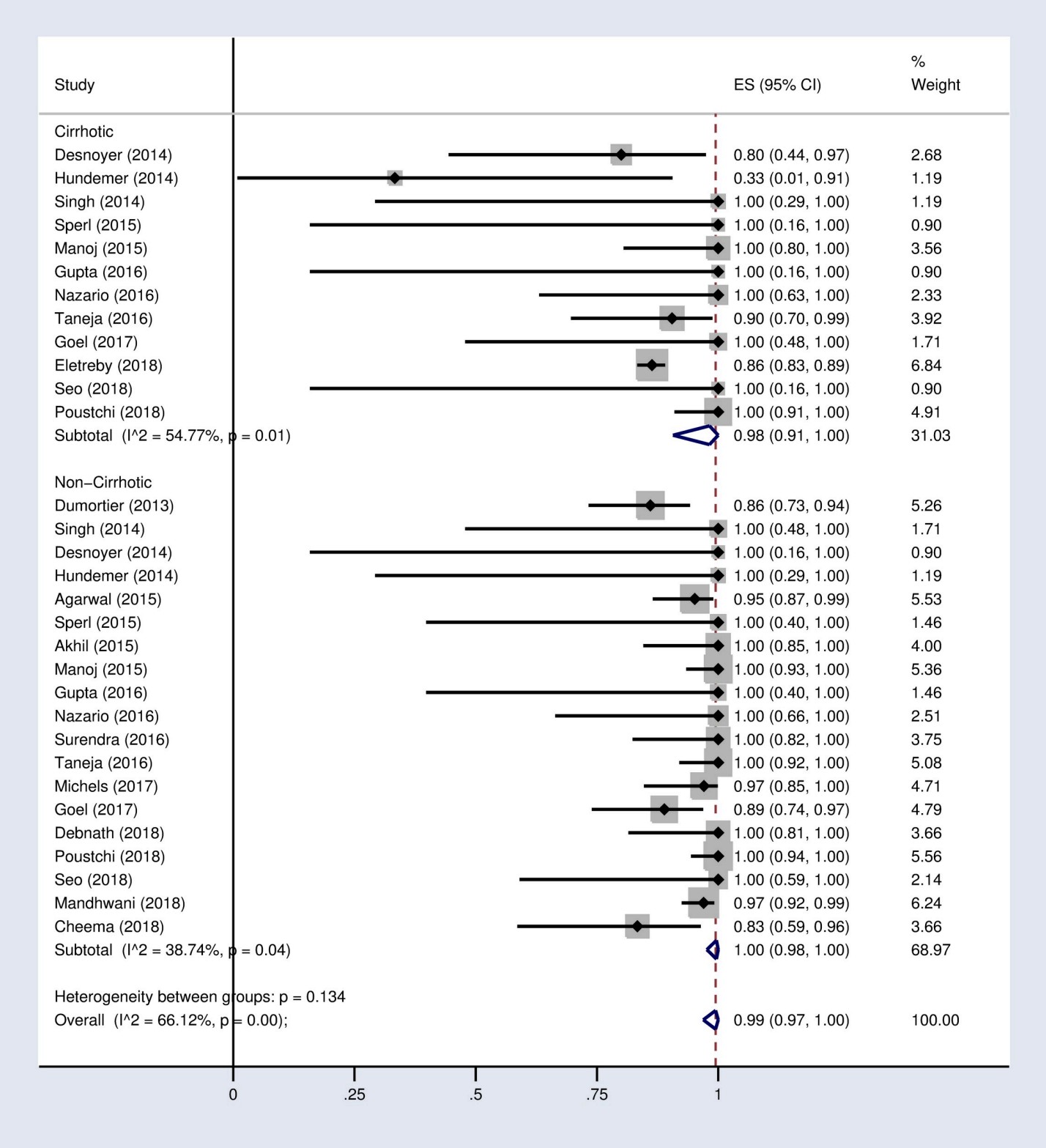

**Fig 3. Forest plot of the pooled SVR12 rate in HCV-infected patients with advanced CKD sub-grouped based on diagnosis of cirrhosis.**

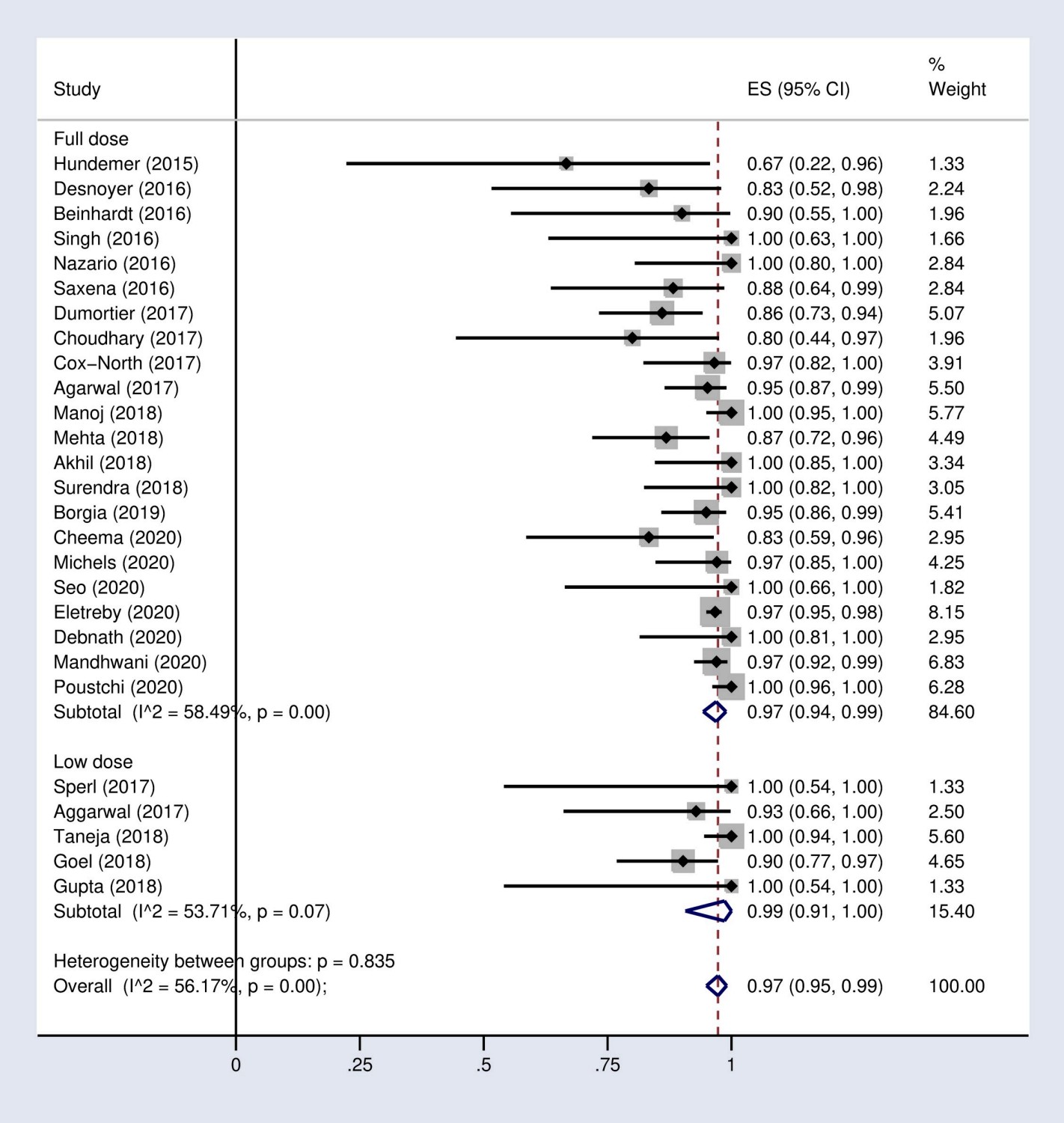

**Fig 4. Forest plot of the pooled SVR12 rate in HCV-infected patient with advanced CKD sub-grouped by SOF-based regimen dose.**

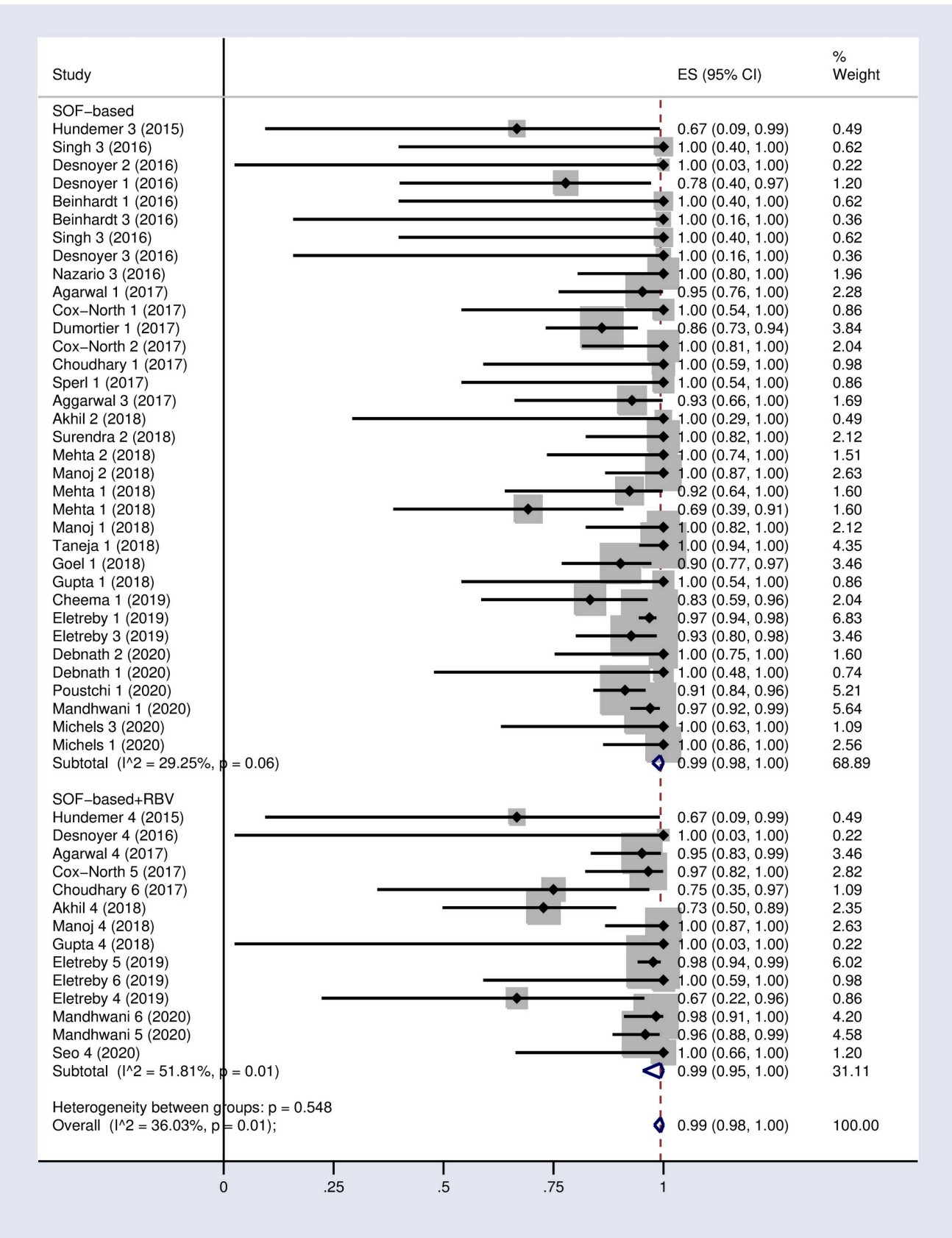

**Fig 5. Forest plot of the pooled SVR12 rate in HCV-infected patients with advanced CKD sub-grouped based on the use of SOF-based regimen with or without RBV.** (1): SOF+DCV; (2): SOF+LDV; (3): SOF+SMV; (4): SOF+RBV; (5): SOF+DCV+RBV; (6): SOF+PEG+RBV.DCV = Daclatasvir. LDV = Ledipasvir. SMV = Simeprevir. RBV = Ribavirin.

SOF-based therapy. The discontinuation rate due to adverse events in the meta-analysis conducted by Li T [51] was 2.2%, which is compatible with our findings.

The pooled incidence of SAE due to SOF-based regimen in our study was 11%, which was similar to the study by Li T et al, in which the pooled SAE rate for DAA-based antiviral therapies in HCV/Stage 4–5 CKD patients was 12.1% [51]. Shehadeh et al also reported a 10% incidence for SAE, which is quite similar to our results [20].

Out of the 15 studies that reported anemia as an adverse event, RBV was used in combination with SOF-based therapy in 11 studies. Also in the meta-analysis conducted by Li T et al in 2016 [51], the most frequent adverse event was anemia, and in the systematic review by Li, M et al in 2019 [50], anemia was the most frequently reported adverse event and RBV was included in almost all of these studies. Shehadeh et al reported fatigue as the most frequent adverse event, followed by anemia. They consistently reported more prevalent anemia in treatment regimens containing RBV. Therefore, regimens containing RBV should be used with caution compared to RBV-free regimens due to high risk of anemia. In the study conducted by Manoj et al, 65.4% of patients who used SOF with RBV developed anemia [26].

Our subgroup analysis suggests that half dose SOF-based regimens are as effective as full dose regimens. The SVR12 rates are similar between half dose (99%) and full dose (97%) regimens. Furthermore, the SAE of half dose regimen (6%) is lower than the full dose treatment (14%). Similarly, Li M et al. (2019) found that both half dose and full dose regimens had considerably high SVR rate (97.1% vs. 96.2%). Li M et al (2019) also suggested lower dose of SOF as routine or usage of half dose once every 2 days with the same efficacy [50]. Shehadeh et al also showed higher efficacy with low-dose treatment [20].

The result of our meta-analysis also showed that the rate of SVR12 in patients who received Sofosbuvir-based treatment without RBV was 99%, which was similar to patients who received SOF with RBV (99%). As we mentioned previously, use of SOF-based treatment in combination with RBV should be used with caution since in addition to increasing the risk of anemia, it increases the discontinuation rate.

Additionally, in cirrhotic patients, the SVR12 rate was slightly lower than non-cirrhotic patients (98% vs. 100%). This finding of our study was similar to the results reported by Rezaee'i-Zavare et al. [52] They recommended that cirrhotic patients be treated with SOF-based regimen in combination with RBV for 12 weeks or without RBV for 24 weeks. Similar to the result of our study, they found that treatment of non-cirrhotic patient with SOF-based regimen without RBV for 12 weeks had acceptable outcomes [52].

Our study is a comprehensive systematic review and meta-analysis on efficacy of SOF-based therapies. We conducted subgroup analysis by combinations of Sofosbuvir with other DAAs regimens and we found out that this regimen is effective and safe in HCV-infected patients with advanced CKD. In addition, these combinations of SOF-based therapies with DAA regimens reduced the incidence of anemia caused by RBV [51].

Our study has certain limitations as well. Firstly, we observed a substantial heterogeneity that might be due to different sampling frames and sample size or different treatment strategies used in a variety of contexts. All included studies were observational without proper control groups. Additionally, almost all studies were conducted in the USA and India. Therefore, the findings of our study cannot be generalized to all populations.

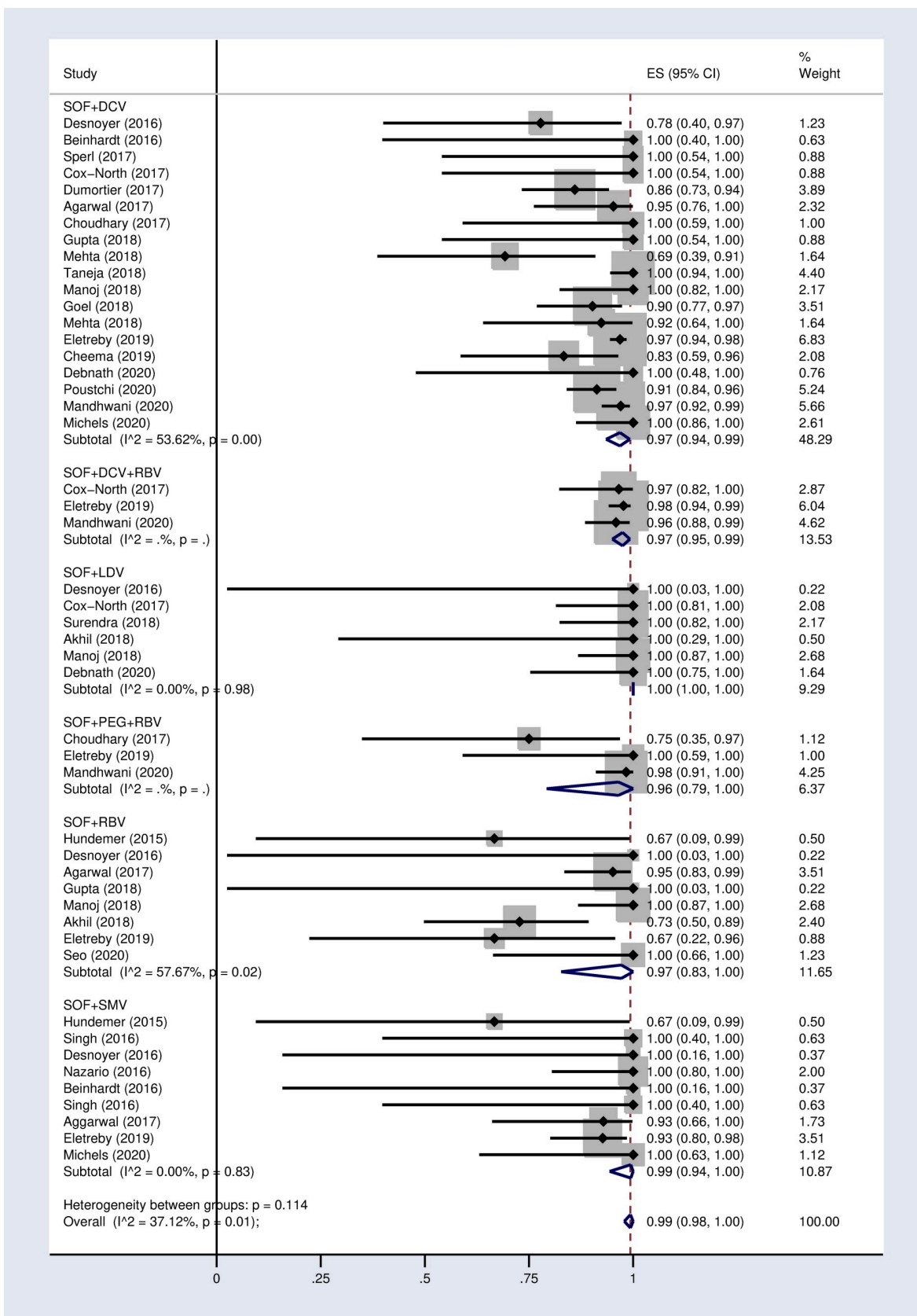

**Fig 6. Forest plot of the pooled SVR12 rate in HCV-infected patients with advanced CKD by treatment strategy.** DCV = Daclatasvir. LDV = Ledipasvir. SMV = Simeprevir. RBV = Ribavirin.

**Table 3. Meta-regression analysis for the effect of suspected variables on the pooled SVR12 rate in HCV-infected patients with advanced chronic kidney disease.**

| Variable | | Univariable Model | | | Multivariable Model | |
|---|---|---|---|---|---|---|
| | β | SE | P Value | β | SE | P Value |
| Age | 18.1 | 7.4 | 0.02 | 10.1 | 9.7 | 0.30 |
| Country | -19.3 | 7.5 | 0.01 | -12.6 | 9.9 | 0.21 |
| Cirrhosis diagnosis | 6.1 | 6.4 | 0.34 | 3.2 | 5.3 | 0.55 |
| treatment strategy | -4.2 | 9.5 | 0.65 | -3.1 | 10.1 | 0.75 |
| dose of drugs | 4.4 | 9.7 | 0.64 | 3.4 | 10.3 | 0.74 |

## Conclusions

The results of this study showed satisfactory and novel findings regarding the usage of half dose SOF-based regimen in HCV-infected patients with advanced CKD. We demonstrated that the use of SOF-based therapy in full or low dose has considerable efficacy in cirrhotic and non-cirrhotic patients. The results of our study have important policy implications as well. The massive production of Sofosbuvir can be a cost-effective strategy to eliminate chronic HCV or at least prevent premature death in advanced CKD patients infected with hepatitis C virus.

## Supporting information

**S1 Checklist. PRISMA 2009 checklist.**
(DOC)

**S1 File.**
(PDF)

## Author Contributions

**Conceptualization:** Sara Majd Jabbari, Sadaf G. Sepanlou.

**Data curation:** Sara Majd Jabbari, Khadije Maajani.

**Formal analysis:** Khadije Maajani, Sadaf G. Sepanlou.

**Supervision:** Sadaf G. Sepanlou.

**Writing – original draft:** Sara Majd Jabbari, Khadije Maajani, Shahin Merat.

**Writing – review & editing:** Shahin Merat, Hossein Poustchi, Sadaf G. Sepanlou.

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
