## [Decision Letter · Decision Letter 0]

3 Nov 2020

PONE-D-20-30679

An updated systematic review and meta-analysis on effectiveness of Sofosbuvir in treating hepatitis C infected patients with end‑stage renal disease

PLOS ONE

Dear Dr. Sepanlou,

Thank you for submitting your manuscript to PLOS ONE. After careful consideration, we feel that it has merit but does not fully meet PLOS ONE’s publication criteria as it currently stands. Therefore, we invite you to submit a revised version of the manuscript that addresses the points raised during the review process.

We look forward to receiving your revised manuscript.

Kind regards,

Chen-Hua Liu

Academic Editor

PLOS ONE

2. Please include the following publication in your introduction and discuss what your research contributes in light of this recent published work: https://www.ncbi.nlm.nih.gov/pmc/articles/PMC7459301/

3. In the Methods, please state the justification the restriction of the timeline to January 2013 onwards.

Reviewers' comments:

Reviewer's Responses to Questions

**Comments to the Author**

1. Is the manuscript technically sound, and do the data support the conclusions?

Reviewer #1: Partly

Reviewer #2: Yes

2. Has the statistical analysis been performed appropriately and rigorously? 

Reviewer #1: Yes

Reviewer #2: I Don't Know

3. Have the authors made all data underlying the findings in their manuscript fully available?

Reviewer #1: Yes

Reviewer #2: Yes

4. Is the manuscript presented in an intelligible fashion and written in standard English?

Reviewer #1: Yes

Reviewer #2: Yes

5. Review Comments to the Author

Reviewer #1: Dear Dr. Sepanlou,

Thank you for your effort on the field about effectiveness and safety of sofosbuvir-based regimen in ESRD patients. Several questions should be clarified.

1. Based on our knowledge, ESRD is defined as patient's eGFR less than 15 ml/min or receiving renal replacement therapy. However, in this study, ESRD was defined as eGFR less than 30 ml/min. You may revise the title and manuscript to "advanced CKD stage" rather than ESRD alone.

2. Line 142, the total patient number is 1571 less than the sum of male and female patients. There would be a mistake in recording patient numbers.

3. Table 1. reference number [26] and [27], the patients were acute infected by HCV virus and the treatment duration was different. Dose these two studies be suitable for enrollment ?

4. Table 1. reference number [30], the eGFR of these 73 renal-impaired patients was less than 45 ml/min. Dose these patient met the inclusion criteria ?

5. Table 1. reference number [31], the SVR of patients with renal impairment is 96.7% but the SVR of patients with ESRD is 80%. However, the SVR of ESRD patients seems not be enrolled in this meta-analysis

6. Table 1, reference number [34], the patient numbers and percentage of cirrhosis were not compatible with original

study, please confirm the accuracy of data extraction

7.please reconfirm the accuracy of patient numbers, SVR rate, the definition of CKD in all the enrolled studies.

Reviewer #2: The authors did an updated systemic review and meta-analysis on the effectiveness of sofosbuvir (SOF)-based regimens in treating hepatitis C virus (HCV)-infected patients with end-stage renal disease (ESRD) from 29 published articles including 1571 subjects. They found that the pooled sustained virologic response (SVR) rates were 97% and 98% 12 and 24 weeks after cessation of antiviral therapies, respectively. Age, cirrhosis, and dose-reduction of SOF did not affect the treatment efficacy while adding on ribavirin (RBV) was associated with a higher incidence of treatment discontinuation and adverse events especially anemia. Though not a novel study, this manuscript has its scientific value in providing healthcare provider guidance in treating chronic HCV-infected patients who have end-stage renal disease. However, several issues deserve clarification or amendment in its current version.

1. Give necessary references to the following sentence “In earlier studies, PEGylated interferon (Peg-IFN) monotherapy…in HCV-infected ESRD patients.” in the second paragraph of the introduction.

2. Change ribavirin abbreviation to “RBV” instead of “rib” in the context.

3. Did HBV co-infection exclude from the enrolled studies?

4. Explain why the number of included studies was greater than 29 in the subgroup analysis table 2?

5. Polish the language for the context such as correct “Sustained viralogical response” to “Sustained virologic response” in Table 2 footnote, and remove unnecessary capital to medication names.

6. In ‘Sensitivity analysis’ please explain why references 30 and 31 were excluded for SVR calculation. The 2nd & 3rd sentences under this subject “The highest and lowest SVR12…respectively. So the pooled SVR12 rate was stable across studies.” were poorly understood.

7. Address the causes of the high incidence of SAE in SOF-based regimen to HCV-infected ESRD patients since the potential harm of its inactive metabolite GS-331007 to these subjects is the most concern issue by healthcare providers.

8. Give proper author names and published year in Figure 2.

6. PLOS authors have the option to publish the peer review history of their article (what does this mean?). If published, this will include your full peer review and any attached files.

Reviewer #1: No

Reviewer #2: No

---

## [Author Response · Author response to Decision Letter 0]

14 Dec 2020

Dear Dr. Chen-Hua Liu,

Thank you very much indeed for considering our manuscript for publication in PLOS One and your detailed comments. We entirely repeated the analyses based on the comments of respected reviewers. Please find our point-by-point response to your comments and the comments of the respected reviewers.

Response: We modified our manuscript according to the requirements of the journal.

2. Please include the following publication in your introduction and discuss what your research contributes in light of this recent published work: https://www.ncbi.nlm.nih.gov/pmc/articles/PMC7459301/

Response: Thanks for this very valuable comment. We included this publication and added the following paragraph to the introduction section of the manuscript:

“In a recent systematic review, the safety and efficacy of Sofosbuvir has been explored among patients on HD.[20] The results are quite similar to the results of the current study. However, there is a very important difference between the two studies. In our study, we included patients with eGFR<30 ml/min/1.73 m2 and our search was not confined to patients on HD. This difference is quite important as the risk of accumulation of toxic metabolites in advanced CKD may be even higher than patients on HD. We also excluded studies on acute HCV and studies on patients with HBV and HIV co-infection. We have also conducted subgroup and sensitivity analyses”.

3. In the Methods, please state the justification the restriction of the timeline to January 2013 onwards.

Response: We mentioned in the methods section that we restricted the timeline to January 2013 as Sofosbuvir was approved for medical use in that year. 

Reviewer comments:

Reviewer #1: Dear Dr. Sepanlou,

Thank you for your effort on the field about effectiveness and safety of sofosbuvir-based regimen in ESRD patients. Several questions should be clarified.

1. Based on our knowledge, ESRD is defined as patient's eGFR less than 15 ml/min or receiving renal replacement therapy. However, in this study, ESRD was defined as eGFR less than 30 ml/min. You may revise the title and manuscript to "advanced CKD stage" rather than ESRD alone.

Response: Thanks for this valuable comment. We replaced ESRD with “advanced CKD” in the title and throughout the manuscript.

2. Line 142, the total patient number is 1571 less than the sum of male and female patients. There would be a mistake in recording patient numbers.

Response: We repeated all analyses based on your valuable comments. We corrected the numbers: “The total sample size of 27 included articles amounted to 1464 patients, among whom 809 were males and 655 were females.” 

3. Table 1. reference number [26] and [27], the patients were acute infected by HCV virus and the treatment duration was different. Dose these two studies be suitable for enrollment?

Response: Thanks for this valuable comment. We dropped these two studies on acute HCV patients and repeated all analyses.

4. Table 1. reference number [30] Saxena, the eGFR of these 73 renal-impaired patients was less than 45 ml/min. Dose these patient met the inclusion criteria?

Response: Thanks for this very valuable comment. You are quite right and we are sorry for this mistake. The study by Saxena et al is mainly conducted on 73 patients with GFR <45 ml/min. But in their subgroup analyses, they have reported the SVR among 17 patients with GFR < 30 ml/min as well. But details about these 17 patients are not reported. Therefore, we used this study only in estimating the overall SVR 12 reported for 17 patients but we dropped it from all other sub-group analyses. 

5. Table 1. reference number [31] Eletreby, the SVR of patients with renal impairment is 96.7% but the SVR of patients with ESRD is 80%. However, the SVR of ESRD patients seems not be enrolled in this meta-analysis 80% is patients on dialysis.

Response: We used the SVR of 96.7% among patients with GFR <30 ml/min in our meta-analysis. We haven’t done a meta-analysis exclusively on patients on hemodialysis.

6. Table 1, reference number [34] Manoj, the patient numbers and percentage of cirrhosis were not compatible with original study, please confirm the accuracy of data extraction.

Response: Thanks very much indeed for this really valuable comment. We corrected the entire table 1.

7. Please reconfirm the accuracy of patient numbers, SVR rate, the definition of CKD in all the enrolled studies.

Response: We reviewed and revised the entire table 1.

Reviewer #2: The authors did an updated systemic review and meta-analysis on the effectiveness of sofosbuvir (SOF)-based regimens in treating hepatitis C virus (HCV)-infected patients with end-stage renal disease (ESRD) from 29 published articles including 1571 subjects. They found that the pooled sustained virologic response (SVR) rates were 97% and 98% 12 and 24 weeks after cessation of antiviral therapies, respectively. Age, cirrhosis, and dose-reduction of SOF did not affect the treatment efficacy while adding on ribavirin (RBV) was associated with a higher incidence of treatment discontinuation and adverse events especially anemia. Though not a novel study, this manuscript has its scientific value in providing healthcare provider guidance in treating chronic HCV-infected patients who have end-stage renal disease. However, several issues deserve clarification or amendment in its current version.

1. Give necessary references to the following sentence “In earlier studies, PEGylated interferon (Peg-IFN) monotherapy…in HCV-infected ESRD patients.” in the second paragraph of the introduction.

Response: Thanks indeed for this valuable comment. We added the following reference:

“KDIGO clinical practice guidelines for the prevention, diagnosis, evaluation, and treatment of hepatitis C in chronic kidney disease. Kidney international Supplement. 2008;(109):S1-99. Epub 2008/05/03. doi: 10.1038/ki.2008.81. PubMed PMID: 18382440.”

2. Change ribavirin abbreviation to “RBV” instead of “rib” in the context.

Response: We used “RBV” throughout the manuscript as you kindly mentioned.

3. Did HBV co-infection exclude from the enrolled studies?

Response: Yes, we excluded studies on patients with HBV or HIV co-infection and we mentioned it in introduction and methods sections.

4. Explain why the number of included studies was greater than 29 in the subgroup analysis table 2?

Response: The number of included studies was greater than 29 (currently 27 studies as we dropped two studies on patients with acute HCV) in subgroups because in some studies, more than one subgroup analysis has been reported. For example, in the study by Desnoyer et al, the SVR for treatment strategies SOF+DCV, SOF+LDV, SOF+RBV, and SOF+SMV have been separately reported. So the results of the study have been reported in all subgroups. Another example is the SVR reported based on whether the treatment was only SOF based or was combined with RBV. In a number of studies, both of these two regimens have been separately reported. 

5. Polish the language for the context such as correct “Sustained viralogical response” to “Sustained virologic response” in Table 2 footnote, and remove unnecessary capital to medication names.

Response: Thanks indeed for this comment. We changed the “Sustained viralogical response” to “Sustained virologic response” throughout the manuscript. 

6. In ‘Sensitivity analysis’ please explain why references 30 and 31 were excluded for SVR calculation. The 2nd & 3rd sentences under this subject “The highest and lowest SVR12…respectively. So the pooled SVR12 rate was stable across studies.” were poorly understood.

Response: As we dropped two studies on acute HCV patients based on the comments of reviewer #1, there are 27 studies in this meta-analysis. In the sensitivity analysis we excluded studies one by one to estimate the effect of their exclusion on the pooled analysis on the remaining 26 studies. When we excluded the study of Durmontier et al (in the revised version of this manuscript), the pooled SVR on the remaining 26 studies was highest (98.9%) and when we excluded the study by Poustchi et al, the pooled SVR on the remaining 26 studies was lowest (95.3%). When we excluded other studies one by one, the SVR on remaining 26 studies lied between 95.3% and 98.9%. We hope this explanation is adequate.

7. Address the causes of the high incidence of SAE in SOF-based regimen to HCV-infected ESRD patients since the potential harm of its inactive metabolite GS-331007 to these subjects is the most concern issue by healthcare providers.

Response: Thanks for this comment. The main message of this study is the low SAE in SOF-based regimens which confirms the utility, the safety, and the effectiveness of this treatment strategy and can’t be a concern for healthcare providers. This point is mentioned in the manuscript.

8. Give proper author names and published year in Figure 2.

Response: Thanks for this comment. All figures were re-drawn and corrected.

---

## [Decision Letter · Decision Letter 1]

31 Dec 2020

PONE-D-20-30679R1

An updated systematic review and meta-analysis on effectiveness of Sofosbuvir in treating hepatitis C infected patients with advanced chronic kidney disease

PLOS ONE

Dear Dr. Sepanlou,

Thank you for submitting your manuscript to PLOS ONE. After careful consideration, we feel that it has merit but does not fully meet PLOS ONE’s publication criteria as it currently stands. Therefore, we invite you to submit a revised version of the manuscript that addresses the points raised during the review process.

We look forward to receiving your revised manuscript.

Kind regards,

Chen-Hua Liu

Academic Editor

PLOS ONE

Reviewers' comments:

Reviewer's Responses to Questions

**Comments to the Author**

1. If the authors have adequately addressed your comments raised in a previous round of review and you feel that this manuscript is now acceptable for publication, you may indicate that here to bypass the “Comments to the Author” section, enter your conflict of interest statement in the “Confidential to Editor” section, and submit your "Accept" recommendation.

Reviewer #1: All comments have been addressed

Reviewer #2: All comments have been addressed

2. Is the manuscript technically sound, and do the data support the conclusions?

Reviewer #1: Partly

Reviewer #2: Yes

3. Has the statistical analysis been performed appropriately and rigorously? 

Reviewer #1: I Don't Know

Reviewer #2: Yes

4. Have the authors made all data underlying the findings in their manuscript fully available?

Reviewer #1: Yes

Reviewer #2: Yes

5. Is the manuscript presented in an intelligible fashion and written in standard English?

Reviewer #1: No

Reviewer #2: Yes

6. Review Comments to the Author

Reviewer #1: Dear Dr. Sepanlou,

Thank you for revised the manuscript. Several issues should be clarified.

1. Line 101 ~ 107: Please revise English. Please do not use "Inclusion criteria :" For example : "All studies published in full English reporting the effectiveness and safety profiles of sofosbuvir-based regimen for treatment-naive chronic HCV-infected patients with advanced chronic kidney disease would be eligible." This would be more fluent and easy to be understood.

2.Line 168~171, Please well clarify the result of your subgroup analysis rather than using the Fig demonstrates ....

3.Line 197~200, Please clarify what you want to tell us. You want to illustrate age and country were associated with SVR12 rate in univariate analysis and none were found to be associated with SVR12 in multi-variate analysis ?

4.Sensitivity analysis : I still could not understand what this mean. Is there any statistical evidence that sensitivity test is done by excluding the lowest and highest extreme values of enrolled studies. The result of pooled SVR12 rate remain around 90~98% after excluding the extreme value didn't mean the meta-analysis is more reliable.

5. English editing of this manuscript would be necessary and helpful.

Reviewer #2: The authors revised the manuscript according to my comments point-by-point, I have no further comments.

7. PLOS authors have the option to publish the peer review history of their article (what does this mean?). If published, this will include your full peer review and any attached files.

Reviewer #1: No

Reviewer #2: No

---

## [Author Response · Author response to Decision Letter 1]

7 Jan 2021

Dear Dr. Chen-Hua Liu,

Thank you very much indeed for considering our manuscript for publication in PLOS One and your detailed comments. Please find our point-by-point response to the comments of the respected reviewers.

Reviewer #1: Dear Dr. Sepanlou,

Thank you for revised the manuscript. Several issues should be clarified.

1. Line 101 ~ 107: Please revise English. Please do not use "Inclusion criteria :" For example : "All studies published in full English reporting the effectiveness and safety profiles of sofosbuvir-based regimen for treatment-naive chronic HCV-infected patients with advanced chronic kidney disease would be eligible." This would be more fluent and easy to be understood.

Response: Thanks indeed for this comment. We modified the text accordingly and used the sentence that you kindly suggested, though the sentence is rather long.

2.Line 168~171, Please well clarify the result of your subgroup analysis rather than using the Fig demonstrates ...

Response: The following sections were added:

“Results sub-grouped by region of study are presented in S3 Fig showing lower SVR12 rates in Europe [95% (89-100)]. Fig 5 demonstrates the pooled SVR12 rate sub-grouped based on being treated only by Sofobuvir [99% (98-100)] or by its combination with RBV [99% (95-100)]. We additionally defined subgroups by treatment strategy such as the result of pooled SVR12 rate in 19 studies in which Sofosbuvir was used in combination with Daclatasvir [97% (94-99)], Simeprevir [99% (94-100)], and Ledipasvir [100% (100-100)], which are reported in Fig 6. Table 2 demonstrates the summary of all sub-group analyses.”

3.Line 197~200, Please clarify what you want to tell us. You want to illustrate age and country were associated with SVR12 rate in univariate analysis and none were found to be associated with SVR12 in multi-variate analysis?

Response: Yes, we want to say the exact points you mention here. We corrected the sentence accordingly:

“There was a significant association between age (P=0.03, β=13.4) and country (P=0.02, β=-18.1) and the pooled SVR12 rate in the univariable model, which disappeared in the multivariable model. None of the variables of cirrhosis diagnosis, treatment strategy, and dose of treatment had significant association with the pooled SVR12 rate, neither in the univariable model nor in the multivariable meta-regression model.”

4.Sensitivity analysis: I still could not understand what this mean. Is there any statistical evidence that sensitivity test is done by excluding the lowest and highest extreme values of enrolled studies. The result of pooled SVR12 rate remain around 90~98% after excluding the extreme value didn't mean the meta-analysis is more reliable.

Response: Sensitivity analyses are integral components of meta-analysis. Sensitivity analysis may explore the impact of excluding or including studies in meta-analysis based on sample size or variance (and not extreme values). If results remain consistent across different analyses, the results can be considered robust as even with different decisions to include or exclude the studies, they remain the same / similar. If the results differ across sensitivity analyses, this is an indication that the result may need to be interpreted with caution. It is very important to note that we haven’t excluded the extreme values. You can notice that Dumortier and Poustchi have not reported extreme values. We excluded each one of the studies out of the entire 27 studies in each step and performed the analyses on the remaining 26 studies. So we repeated 27 sensitivity analyses and we made 27 different estimates of SVR12 upon exclusion of each study from the list. The highest and the lowest estimates across these 27 analyses were reported in the text upon exclusion of studies by Dumortier and Poustchi respectively. Here, the analyses show that the results of these 27 sensitivity analyses are consistent and within the range of the overall SVR12 across all 27 studies. Therefore, our results are actually robust and don’t depend on the choice of included studies. We hope these explanations are adequate. 

We added the following section to the manuscript:

“To assess the effect of all studies on the pooled SVR12 rate, we used sensitivity analyses. In each step, we excluded one of the total of 27 studies and calculated the pooled SVR12 rate across the remaining 26 studies. Thus, we performed 27 sensitivity analyses and we made 27 different estimates of SVR12 upon exclusion of each study in each step. The highest and lowest estimates after excluding the studies by Dumortier [28] and Poustchi [43] were 98.9% (90.8-100) and 95.3% (91.1-100) respectively. So the pooled SVR12 rate was stable across studies. Our results are actually robust and don’t depend on the choice of included studies.” 

5. English editing of this manuscript would be necessary and helpful.

Response: Thanks for this very important comment. We edited the entire manuscript. We made a major change and used the term “efficacy” instead of “effectiveness” in the title and throughout the entire text of the manuscript.

Reviewer #2: The authors revised the manuscript according to my comments point-by-point, I have no further comments.

Response: Thanks indeed for your positive feedback.

---

## [Decision Letter · Decision Letter 2]

22 Jan 2021

An updated systematic review and meta-analysis on efficacy of Sofosbuvir in treating hepatitis C-infected patients with advanced chronic kidney disease

PONE-D-20-30679R2

Dear Dr. Sepanlou,

We’re pleased to inform you that your manuscript has been judged scientifically suitable for publication and will be formally accepted for publication once it meets all outstanding technical requirements.

Kind regards,

Chen-Hua Liu

Academic Editor

PLOS ONE

Reviewers' comments:

Reviewer's Responses to Questions

**Comments to the Author**

1. If the authors have adequately addressed your comments raised in a previous round of review and you feel that this manuscript is now acceptable for publication, you may indicate that here to bypass the “Comments to the Author” section, enter your conflict of interest statement in the “Confidential to Editor” section, and submit your "Accept" recommendation.

Reviewer #1: All comments have been addressed

2. Is the manuscript technically sound, and do the data support the conclusions?

Reviewer #1: Yes

3. Has the statistical analysis been performed appropriately and rigorously? 

Reviewer #1: Yes

4. Have the authors made all data underlying the findings in their manuscript fully available?

Reviewer #1: Yes

5. Is the manuscript presented in an intelligible fashion and written in standard English?

Reviewer #1: Yes

6. Review Comments to the Author

Reviewer #1: Dear Dr. Sepanlou,

Thank you to revised the manuscript according to my comment point-to-point and made English editing. I do not have further comment on this study.

7. PLOS authors have the option to publish the peer review history of their article (what does this mean?). If published, this will include your full peer review and any attached files.

Reviewer #1: No

---

## [Editor Report · Acceptance letter]

28 Jan 2021

PONE-D-20-30679R2 

An updated systematic review and meta-analysis on efficacy of Sofosbuvir in treating hepatitis C-infected patients with advanced chronic kidney disease 

Dear Dr. Sepanlou:

I'm pleased to inform you that your manuscript has been deemed suitable for publication in PLOS ONE. Congratulations! Your manuscript is now with our production department. 

Kind regards, 

on behalf of

Dr. Chen-Hua Liu 

Academic Editor

PLOS ONE